

# Do Tonkean macaques (*Macaca tonkeana*) perceive what conspecifics do and do not see?

Charlotte Canteloup[1,2,3], Emilie Piraux[2], Nicolas Poulin[4] and Hélène Meunier[1,2,3]

[1] Laboratoire de Neurosciences Cognitives et Adaptatives, UMR 7364, Strasbourg, France
[2] Centre de Primatologie de l'Université de Strasbourg, Niederhausbergen, France
[3] University of Strasbourg Institute for Advanced Study, Strasbourg, France
[4] Centre Statistique de Strasbourg, IRMA, UMR 7501, Strasbourg, France

Corresponding author
Charlotte Canteloup,
charlotte.canteloup@gmail.com

## ABSTRACT

The understanding of the visual perception of others, also named visual perspective taking, is a component of Theory of Mind. Although strong evidence of visual perspective taking has been reported in great apes, the issue is more open to discussion in monkeys. We investigated whether Tonkean macaques (*Macaca tonkeana*) know what conspecifics do and do not see, using a food competition paradigm originally developed in great apes. We tested individuals in pairs, after establishing the dominance relationship within each pair. Twenty-one pairs were tested in four different conditions. In one condition, the subordinate had the choice between two pieces of food, one that was visible only to it and another that was also visible to the dominant. It was predicted that if the subordinate understands that the dominant cannot see both pieces of food because one is hidden from its view, the subordinate should preferentially go for the food visible only to itself. In the three other conditions, we varied the temporal and visual access to food for both individuals, to control for alternative explanations based on dominance. We recorded the first movement direction chosen by subjects, i.e. towards a) visible food b) hidden food or c) elsewhere; and the outcome of the test, i.e. the quantity of food obtained. Results showed that subordinates moved preferentially for the hidden food when released simultaneously with the dominant and also with a head start on the dominant. By contrast, dominants' choices of the two pieces of food were random. We also describe and discuss some of the strategies used by subordinates in these tests. According to the whole of our results, Tonkean macaques seem capable of visual perspective taking despite the fact that a low-level explanation as behavior reading has not been totally excluded.

## INTRODUCTION

Exploiting information from others is an ability that plays a key role in social species' daily interactions. One way of obtaining information is gaze following, defined as "looking where someone else is looking" (*Corkum & Moore, 1995*), a widespread behavior in nonhuman primates (*Emery, 2000*; *Itakura, 2004*). Gaze following may rely

on simple mechanisms and is adaptive for detecting food during foraging, potential danger from predators, and important social information such as dominance-related interactions (*Itakura, 2004*). Moreover, it is considered as a prerequisite to complex forms of social cognition such as Theory of Mind (*Emery, 2000*). The use of another's gaze cues has indeed been implicated in higher order cognitive abilities such as visual perspective taking, deception and empathy (e.g. *Anderson, 1998*; *Tomasello et al., 2005*; *Byrne & Bates, 2010*). In these cases, the observer understands that the observed individual is attending to a particular stimulus because the individual intends to do something with the visual target, or believes something about it. One of the biggest issues faced by comparative cognitive scientists is the extent to which gaze following reflects mental state attribution, especially in our closest relatives, the nonhuman primates (*Shepherd, 2010*).

Following the gaze of a human experimenter has been reported in many primate species (chimpanzees, *Pan troglodytes*: *Povinelli & Eddy, 1996*; *Call, Hare & Tomasello, 1998*; *Tomasello, Hare & Fogleman, 2001*; gibbons, *Hylobates pileatus, H. moloch, H. lar, Symphalangus syndactylus*: *Liebal & Kaminski, 2012*; rhesus macaques, *Macaca mulatta*: *Tomasello, Hare & Fogleman, 2001*; stump-tailed macaques, *Macaca arctoides*: *Anderson & Mitchell, 1999*; marmosets, *Callithrix jacchus*: *Burkart & Heschl, 2006*). Conspecific models are also effective (chimpanzees: *Tomasello, Call & Hare, 1998*; *Kano & Call, 2014*; bonobos, *Pan paniscus*: *Kano & Call, 2014*; orangutans, *Pongo abelii*: *Kano & Call, 2014*; rhesus macaques: *Emery et al., 1997*; *Tomasello, Call & Hare, 1998*; stumptailed macaques: *Tomasello, Call & Hare, 1998*; pigtail macaques, *Macaca nemestrina*: *Tomasello, Call & Hare, 1998*; sooty mangabeys, *Cercocebus atys torquatus*: *Tomasello, Call & Hare, 1998*; brown lemurs, *Eulemur fulvus* and black lemurs, *Eulemur macaco*: *Ruiz et al., 2009*). Furthermore, some scholars have recently reported flexibility in primate gaze following skills. For example, crested macaques (*Macaca nigra*) reacted quicker to a change of gaze direction by a socially close conspecific (*Micheletta & Waller, 2012*), and long-tailed macaques (*Macaca fascicularis*) showed stronger gaze following responses when a human's gaze shift was accompanied by a social facial expression compared to a neutral expression (*Goossens et al., 2008*). The latter authors also showed that macaques frequently looked back to the human's face when facing a human who was looking at the ceiling where there was nothing of interest; chimpanzees do likewise (*Call, Hare & Tomasello, 1998*; *Bräuer, Call & Tomasello, 2005*). Such results suggest that gaze following is more than a simple reflex, but they provide no evidence that great apes understand that the model is actually seeing something. This ability to understand what another subject can see, also named visual perspective taking, is a component of Theory of Mind (*Premack & Woodruff, 1978*; *Povinelli, Nelson & Boysen, 1990*).

The development of visual perspective taking in human infants is a two-step process, according to *Flavell (1992)*. For this author, infants near the age of two years can infer that someone may see something that they do not, and *vice versa* (called Level 1 knowledge of visual perception). Later, around three years of age (e.g. *Moll & Meltzoff, 2011*), infants are able to recreate the different visual appearances of something viewed by two persons from different locations (called Level 2 knowledge of visual perception). Some researchers have

sought to determine whether nonhuman primates are capable of Level 1 perspective taking by testing great apes and monkeys in various experimental paradigms.

One of the most commonly used paradigms to study visual perspective taking in nonhuman primates is the object-choice task. Typically in this task, a human experimenter hides a piece of food under one of two containers and then attempts to notify the subject about the location of the food by staring or pointing at the baited container. Great apes (chimpanzees: *Call, Agnetta & Tomasello, 2000*; orangutans: *Call & Tomasello, 1998*; gorillas: *Peignot & Anderson, 1999*) and monkeys (brown capuchins: *Anderson, Sallaberry & Barbier, 1995*; rhesus macaques: *Anderson, Montant & Schmitt, 1996*; marmosets: *Burkart & Heschl, 2007*, but see *Burkart & Heschl, 2006*) often fail to request food from the correct container when the human's gaze is the only cue. In another experiment (*Povinelli & Eddy, 1996*), chimpanzees that were trained to beg for food were confronted with two experimenters who could potentially give them food: one could see them and one could not. Because the chimpanzees did not solicit food from a human who could see them more than from one who could not see them, the authors concluded that chimpanzees do not have a mentalistic understanding of seeing (i.e. knowing that seeing is "about" something). Instead, according to *Povinelli & Eddy (1996)*, chimpanzees have a behaviorist appreciation of the behavior of others (i.e. "the chimpanzees' behavior is completely governed by observable entities and events without recourse to reasoning about unobservable mediating mental states," footnote 1 p 25).

Some researchers (*Johnson & Karin-D'Arcy, 2006* for review) have criticized the object-choice task as it diverges greatly from social conditions in which primates naturally use visual perspective taking; in other words it lacks ecological validity. Specifically, the object-choice task puts the subject in a cooperative situation with a human partner, an unusual context unlikely to occur in nonhuman primates' natural daily life. Therefore, acknowledging the importance of competition in primates' normal lives (*Byrne & Whiten, 1988*), researchers have developed alternative paradigms involving an intraspecific competitive context to test visual perspective taking in a more ecologically valid way (*Hare, 2001*). In the pioneering experiment by *Hare et al. (2000)*, two chimpanzees–a dominant and a subordinate–were tested in a food competition situation. The subjects were placed in opposite rooms with food pieces positioned in a room situated centrally between the dominant's room and the subordinate's room. In the test, one piece of food was visible to both individuals while the other, hidden behind a barrier, was visible only to one of the two subjects. *Hare et al. (2000)* hypothesized that when subordinates saw both pieces of food, they would head for the hidden food more often than the visible one, which would be evidence that they could take the visual perspective of their dominant conspecific. Subordinate chimpanzees did indeed preferentially go for the food that only they could see when released simultaneously with the dominant (*Hare et al., 2000*; *Bräuer, Call & Tomasello, 2007*). The same result was obtained when subordinates were released shortly before the dominant, forcing them to make a choice of direction towards one of the two pieces of food before the dominant that not depends on the dominant's intention movements towards food (*Hare et al., 2000*; *Hare, Call & Tomasello, 2001*; *Bräuer, Call & Tomasello, 2007*). However, these authors reported an alternative

hypothesis named the intimidation hypothesis: subordinates could choose the food the dominant was not looking at throughout the space under the trapdoor allowing subject to see the testing area. In order to rule out this hypothesis, Hare and collaborators assessed two controls. A first control was made nine months later in which the dominant's door was raised for some seconds after the baiting procedure and finally closed. Next, the subordinate's door was opened allowing the subordinate to enter in the testing area. The dominant was then released after the subordinate had adopted its first direction choice. Researchers reported that subordinates approached the hidden piece of food more often than the visible one. A second control placed the dominant in a situation in which it could see always a single piece of food hidden from the subordinate's view. In this control, two experimental conditions were performed: in one, both trapdoors were raised allowing both subjects to see the occluders but only the dominant could see the hidden piece of food. In this condition, the subordinate had the opportunity to read the dominant's behavior and locate food whereas in the second condition, only the subordinate's door was raised, so it could not read the dominant's behavior. Hare and collaborators reported that subordinates chose to head for one of the two occluders at random, so they did not use the dominant's behavior to locate food, which strengthen the hypothesis that chimpanzees know what conspecifics do and do not see.

Few comparable studies have been done on monkeys, with equivocal results regarding their visual perspective taking abilities. On one hand, subordinate capuchins (*Sapajus apella*) have been reported to prefer retrieving hidden food when released at the same time as dominants; however, they did not approach hidden food first when released with a short head start. Thus, capuchins appeared to base their choice on behavior reading rather than perspective taking (*Hare et al., 2003*). Marmosets reportedly behaved like chimpanzees (*Burkart & Heschl, 2007*) in conditions similar to those designed by *Hare et al. (2000)* and *Hare et al. (2003)* but the authors also acknowledged a possible behavior reading explanation. On the other hand, some lemur species (ring-tailed lemur, *Lemur catta*: *Sandel, MacLean & Hare, 2011*; *MacLean et al., 2013*; *Bray, Krupenye & Hare, 2014*; black lemur; brown lemur; Coquerel's sifaka, *Propithecus coquereli*: *MacLean et al., 2013*) and rhesus macaques (*Macaca mulatta*: *Flombaum & Santos, 2005*; *Bray, Krupenye & Hare, 2014*) have been shown to preferentially steal food from a human competitor who cannot see the food rather than one who can see it. In addition, *Overduin-de Vries, Spruijt & Sterck (2014)* reported that long-tailed macaques understood what a conspecific competitor could see; those authors ruled out a behavior reading explanation because only subordinates could see the dominants when making their choice, whereas dominants had no visual access to food or the subordinates either before or during the trial. In this situation dominants could not show special interest in one of the two pieces of food, and consequently they had no inhibiting effect on the subordinates. Social cognitive studies in macaques focused essentially on despotic species such as rhesus and long-tailed macaques, neglecting tolerant species as their Sulawesian cousins. How specific social dynamics affect cognitive abilities as Theory of Mind abilities is poorly understood and studies of socio-cognitive capacities tend to overlook the potential influence of social characteristics of the species. Studying socially tolerant species could thus provide insights

about the influence of sociocultural environment on perception reading abilities and can help to elucidate the ecological and social pressures that favoured the evolution of Theory of Mind. Social tolerance being one of the important factors for social complexity, we would expect more complex cognitive skills for a more socially tolerant species compared to a more despotic one. Testing a tolerant species in a competitive context also allows investigation of the contextual features that underlie such cognitive skills in species with different ecological and social characteristics.

In this context, we tested a tolerant macaque species that has not been extensively studied, the Tonkean macaque (*Macaca tonkeana*), in an experiment inspired by *Hare et al. (2000)* and *Hare et al. (2003),* to investigate their understanding of visual perception. For this, we used an equivalent of the "occluder test" from *Hare et al. (2000)* in four different experimental conditions. In conditions 1 and 2, the subordinate had visual access to two pieces of food: one positioned on the top of the visual barrier, visible to the subject and to the dominant, and one positioned under the barrier, thus only visible to the subject. The difference between conditions 1 and 2 was that in condition 1, both monkeys had access to the testing area at the same time, whereas in condition 2 the subordinate was released slightly earlier than the dominant. Thus, condition 2 was a control aimed at ruling out the possibility subordinates might simply react to the dominant's intention movements towards the visible food. In conditions 3 and 4, the dominant had visual access to both pieces of food whereas the subordinate saw only one. Conditions 3 and 4 differed in that in condition 4, both monkeys had access to the testing area at the same time, whereas in condition 3, the dominant was released with a short head start on the subordinate. These two conditions were run to verify that macaques do not have a general preference for hidden food and that dominants do not exhibit specific behavioral strategies in the presence of subordinates. We expected that, as in long-tailed macaques, subordinate Tonkean macaques would move preferentially for the hidden food in conditions 1 and 2, whereas dominants would show no preference for either piece of food in conditions 3 and 4.

## METHODS

### Ethical note

The procedures used here adhere to the French legal requirements for the Use of Animals in Research. This experiment was approved by the Animal Experiment Committee of the Centre de Primatologie de l'Université de Strasbourg and by the CREMEAS Ethics Committee (Approval for conducting experiments on primates n° AL/46/53/02/13).

### Subjects

The subjects were eleven Tonkean macaques (eight males aged two–11 years and three females aged four–17 years), tested in 21 dyads, all born and raised at the Centre de Primatologie de l'Université de Strasbourg, France. Four subjects were tested as dominant only, one as subordinate only and six as both dominant and subordinate (see Table 1 for details). Subjects lived in a social group of 28 individuals in a one-acre wooded park with access to a 20 m$^2$ heated indoor housing area. The monkeys' diet consisted of

**Table 1** Sex, age, hierarchical rank, number of trials as dominant, number of trials as subordinate and total number of trials of subjects.

| Subject | Sex | Age (years) | Hierarchical rank | Number of trials as dominant | Number of trials as subordinate | Total number of trials |
|---------|-----|-------------|-------------------|------------------------------|----------------------------------|------------------------|
| Lady | Female | 17 | 12 | 45 | 166 | 211 |
| Yannick | Male | 5 | 13 | 6 | 197 | 203 |
| Nereis | Female | 15 | 8 | 13 | 7 | 20 |
| Nema | Female | 3 | 14 | 45 | 11 | 56 |
| Vishnu | Male | 8 | 6 | 32 | 20 | 52 |
| Uruk | Male | 9 | 2 | 6 | 0 | 6 |
| Wallace | Male | 7 | 5 | 75 | 0 | 75 |
| Wotan | Male | 7 | 9 | 128 | 4 | 132 |
| Walt | Male | 7 | 10 | 54 | 0 | 51 |
| Shan | Male | 11 | 1 | 31 | 0 | 31 |
| Nenno | Male | 2 | 19 | 0 | 30 | 30 |

commercial pellets and water *ad libitum*, and fruits and vegetables twice a week, after experimental sessions.

## Apparatus

Subjects were tested in dyads in an outdoor experimental enclosure adjacent to their park, which allowed individuals to participate voluntarily. The experimental enclosure consisted of three compartments interconnected by trapdoors (Fig. 1): an area "A" (430 × 290 cm and 190 cm high) directly connected to the park; a middle compartment "B" or testing area (290 × 290 cm and 190 cm high), and an area "C" (140 × 290 cm and 190 cm high). In compartment B two breeze blocks (49 × 38 cm and 15 cm high), served as visual barriers; they were placed in the middle of the area, equidistant to the trapdoors and separated by 80 cm. The three compartments were connected by double-layered trapdoors (42 × 52 cm) consisting of one transparent Plexiglas door doubled with an opaque metal door. The two layers of the trapdoors could be opened independently.

## Establishment of dominance hierarchy within the group

The first step in this study was to determine dominance relationships between the subjects. This was done in two ways: firstly, by establishing the dominance hierarchy within the group, and secondly, verifying the dominance relationship within tested dyads.

The within-group dominance hierarchy was determined in 25 sessions of food competition tests. A bottle filled with diluted sweet syrup was attached to the wire mesh fencing inside the macaques' park. This induced food competition for access to the syrup, and the experimenter recorded agonistic interactions in the vicinity of the bottle. At the end of each interaction, each participant was recorded as "winner" or "loser." From the resulting global matrix the dominance hierarchy was obtained using Matman 1.1 (*De Vries, Netto & Hanegraaf, 1993*); it was significantly linear ($h' = 0.39$; $P = 0.00009$). Then we calculated hierarchical rank differences for each dyad to be tested

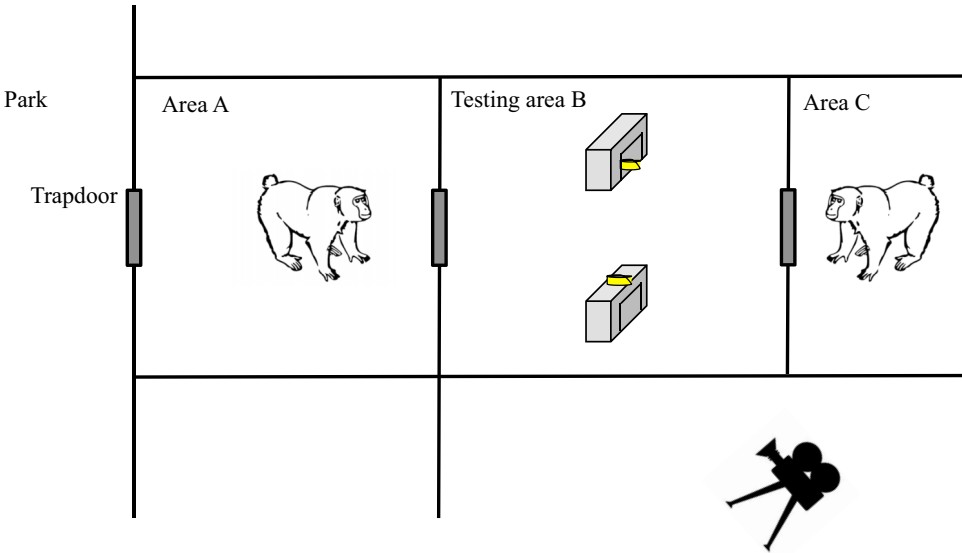

**Figure 1** Schema of the experimental apparatus used in the visual perspective-taking experiment. Macaques in area A and C were given access to the testing area B where two pieces of food were placed. One piece was put on one of the two breeze blocks, i.e. visible by the two subjects (subjects in A and C), and another one was placed in and under the other breeze block, i.e. hidden from one of the two subject's view (subject in A).

by subtracting the rank of the subordinate from that of the dominant (see raw data for details).

## Establishment of dominance relationship within dyads

Determination of the dominance relationship within dyads took place in the experimental area. Twenty-one dyads were tested between April and June 2014. One subject was placed in area A and the other in the area C (Fig. 1). The opaque trapdoors were lowered to hide the baiting procedure from each monkey. A slice of banana was placed centrally in the testing area, exactly half way between the two breeze blocks. The Experimenter (EP) then simultaneously raised the opaque layers of the two trapdoors to allow the subjects to see into the testing area through the transparent layers. Five seconds after the two subjects were in position behind the Plexiglas trapdoors and after having looked into the testing area, the experimenter simultaneously raised the transparent layers, allowing the monkeys to enter the testing area. Whichever monkey picked up the slice of banana was considered dominant in this trial. We performed 10 such tests per dyad and tested the significance of the difference in successes using a binomial test for each dyad.

## Experimental procedure

The experiment took place in the three experimental areas described above (Fig. 1). For each trial, one slice of banana was placed on top of one of the two breeze blocks, and the other was put inside and near the back of the other breeze block, so that only one of the two subjects could see both banana pieces (subject in C in Fig. 1). Which piece of food

went in which location was randomized across trials. The experiment consisted of four experimental conditions as follows: In conditions 1 and 2, the subordinate could see both pieces of food, but in condition 1 both subordinate and dominant were released into the testing area simultaneously, whereas in condition 2 the subordinate was released slightly before the dominant (the dominant was released as soon as the subordinate's entire body was in the testing area). In conditions 3 and 4 the dominant could see both pieces of food, but in condition 3 the dominant was released slightly before the subordinate, whereas in condition 4 both individuals were released simultaneously. In cases when the individual released first (conditions 2 and 3) did not fully enter the testing area, the other individual was released 60 seconds after the first individual. Because subjects were tested opportunistically (i.e., they had free access to the experimental area and were never forced to participate), we ran between 1 and 22 trials of the four different experimental conditions following a same random list of trials per dyad (see data acquisition sheet on raw data for details), resulting in a total of 1 to 66 trials per dyad (total mean of trials per dyad = 20.7 ± 4.3; see Table 2 for details), with a maximum of twenty trials per condition in total (ten trials for each breeze block). Each trial started when at least one of the two transparent trapdoors was opened, and ended when the two pieces of food were picked up.

## Data and reliability analyses

All experiments were recorded using a video camera HD (Canon, Legria HF S20) and data were later coded by CC using the software VLC 2.1.5. Two variables were measured: a) the first movement direction taken by individuals: (i) hidden: head and chest oriented towards the hidden banana; ii) visible: head and chest oriented towards the visible banana; iii) elsewhere: the subject did not enter in the testing area or entered and sat down in front of the trapdoor or entered and crossed the testing area between the two breeze blocks); b) the outcome of the test: (i) outcome 1: the dominant gets both pieces of food ii) outcome 2: the subordinate gets both pieces of food; iii) outcome 3: the dominant gets the visible piece of food and the subordinate gets the hidden one; iv) outcome 4: the subordinate gets the visible piece of food and the dominant gets the hidden one). In some cases, subjects behaved as if they understood that two pieces of food were present and began searching both locations, even on trials in which they only had visual access to one piece of food. In these trials, just after having retrieved the visible piece of food, some dominants systematically visited the other occluder on the subordinate's side to retrieve the hidden banana. We excluded these trials from the analysis. Twenty-nine trials were removed due to errors during experiments (e.g. one subject was released in the testing area too late). Ten additional trials were removed from the analysis due to the loss of two video clips (trial n°1 for the dyad Wotan-Yannick and trials 4 to 12 for the dyad Shan-Yannick). For reliability analysis, a random twenty percent of the videos were analysed by HM. Inter-observer agreements were excellent for both the first movement direction (Cohen's $\kappa$ = 0.94) and the outcome of the test (Cohen's $\kappa$ = 0.99).

**Table 2** Hierarchical rank difference and number of trials per condition of tested dyads.

| Tested dyad (dominant–subordinate) | Hierarchical rank difference | Number of trials condition 1 | Number of trials condition 2 | Number of trials condition 3 | Number of trials condition 4 |
|---|---|---|---|---|---|
| Wallace–Yannick | 8 | 10 | 8 | 7 | 9 |
| Wotan–Yannick | 4 | 20 | 14 | 11 | 21 |
| Walt–Yannick | 3 | 3 | 5 | 4 | 2 |
| Lady–Yannick | 1 | 3 | 6 | 4 | 2 |
| Nereis–Lady | 4 | 3 | 3 | 3 | 2 |
| Walt–Lady | 2 | 9 | 11 | 9 | 11 |
| Wotan–Lady | 3 | 16 | 18 | 15 | 13 |
| Nema–Lady | 2 | 11 | 13 | 10 | 11 |
| Vishnu–Yannick | 7 | 7 | 8 | 10 | 7 |
| Lady–Nenno | 7 | 7 | 8 | 9 | 6 |
| Shan–Yannick | 12 | 8 | 7 | 6 | 10 |
| Wallace–Wotan | 4 | 0 | 2 | 0 | 0 |
| Wallace–Vishnu | 1 | 3 | 8 | 5 | 4 |
| Uruk–Yannick | 11 | 0 | 2 | 0 | 1 |
| Wallace–Lady | 7 | 0 | 3 | 2 | 2 |
| Yannick–Nema | 1 | 0 | 3 | 1 | 2 |
| Uruk–Wotan | 7 | 0 | 1 | 0 | 1 |
| Nereis–Yannick | 5 | 0 | 1 | 0 | 1 |
| Wallace–Nereis | 3 | 0 | 3 | 2 | 2 |
| Wallace–Nema | 9 | 0 | 1 | 2 | 2 |
| Uruk–Lady | 10 | 0 | 0 | 0 | 1 |

## Statistical analysis

Analysis focused on experimental configurations where subjects chose between the two pieces of food, namely conditions 1 and 2 for subordinates and conditions 3 and 4 for dominants. In order to deal with pseudoreplication, we used several Generalized Linear Mixed Models (GLMMs) with identities of the dominant, the subordinate and the dyad as random effects. Each combination of the random variables was used in different models, and backward selection of the random effects based on Likelihood Ratio Test (LRT; *Lewis, Butler & Gilbert, 2011*) was performed for each model. In each model, experimental condition, trial number and hierarchical rank difference were assessed as fixed effects. We used these models to determine which factors most influenced the subjects' choice of direction and the outcome of the test.

Regarding subject choice, the analyses were divided into two sub-questions: 1) Which factors explain the choice of direction or no choice? 2) When subjects made a choice, which factors explain the direction chosen, i.e. hidden or visible? To answer these questions, GLMMs (GLMM1 to GLM4) for data following a binomial distribution with a logit link function were fitted. Additionally, to investigate potential learning effects on the

first movement direction and for a better comparison with results of previous studies on other species, we analysed only the two first trials per dyad using Wilcoxon signed rank tests.

Concerning the outcome of the test, we fitted a GLMM for each possible outcome (GLMM5 to GLMM8) to investigate whether the outcome depended on the experimental condition. Some outcomes did not occur for some conditions. Hence, we fitted: i) a model for outcome 1 on the basis of the four experimental conditions; ii) a model for outcome 2 on the basis of conditions 1 and 2; iii) a model for outcome 3 on the basis of conditions 1 and 2 and iv) a model for outcome 4 on the basis of conditions 3 and 4. Moreover, to investigate potential learning effects on outcome, and to better compare with the results of previous studies on other species, we analysed only the two first trials per dyad using Wilcoxon matched pairs tests.

To investigate whether subordinates obtained both pieces of food (outcome 2) significantly more often as a function of their first movement direction (i.e. towards the visible or the hidden food), we fitted a GLM (GLM9) including their first direction as a fixed effect.

Tukey corrections were applied when performing multiple comparison tests between experimental conditions. All models were performed using R 3.1.2's package lme4 (*Bates et al., 2014*) and Wilcoxon tests were done with GraphPadInstat 3.1 with alpha set at 0.05.

## RESULTS

### First movement direction: towards food or not?

The best-fitting model on the basis of LRT was the one with the identity of the dominant as the only random effect (GLMM1; see details of each model in Table S1). According to this model, subordinates headed for food rather than elsewhere when they saw both pieces of food and were released at the same time as the dominants (condition 1; $z = 2.61$; $P = 0.009$), and with a head start (condition 2; $z = 5.44$; $P < 0.0001$) on the dominants (Fig. 2). In contrast, they mostly headed elsewhere instead of towards food when they saw only one piece of food and were released after the dominant (condition 3; $z = -2.72$; $P = 0.007$). When they saw only one piece of food and were released at the same time as the dominant, subordinates did not show any preference for approaching food or going elsewhere (condition 4; $z = -0.18$; $P > 0.05$). The probability that the subordinates would first go towards food increased significantly across trials ($z = 2.74$; $P = 0.006$).

We established another model (GLMM2) focusing only on conditions 1 and 2 to compare the behavior of subordinates in these two conditions. The best model on the basis of LRT was the one with only the dyad as random effect. According to this model, subordinates headed for food significantly more than elsewhere when they were released before the dominant than were released at the same time as the dominant ($z = 3.33$; $P = 0.0009$). The greater the hierarchical rank difference, the less likely subordinates headed towards food ($z = -2.26$; $P = 0.02$).

Dominants headed directly for food in all conditions (Fig. 2; see Table S2 for details).

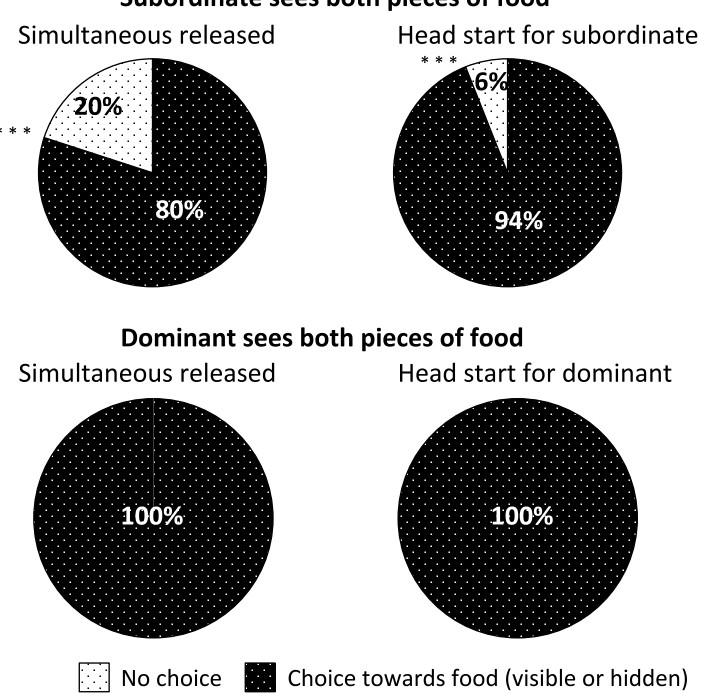

**Figure 2** First direction choice taken by the subject able to see both pieces of food, i.e. the subordinate in the conditions 1 and 2 and the dominant in the conditions 3 and 4, in the different experimental conditions. "No choice" category corresponds to cases where the subject did not enter in the testing area or entered and sat down in front of the trapdoor or entered and crossed the testing area through the two breeze blocks. "Choice towards food" category corresponds to cases where the subject heads for hidden or visible food. *** $P < 0.0001$.

## First direction choice: hidden or visible food?

The best-fitting model for when subordinates headed for food (hidden or visible) concerning conditions 1 and 2 was the one with the identity of the tested dyad as the only random effect (GLMM3). According to this model, subordinates headed for hidden food significantly more than visible food when they were released simultaneously (condition 1; $z = 4.64$; $P < 0.0001$) and with a head start on the dominant (condition 2; $z = 2.87$; $P = 0.004$) (Fig. 3). As the number of trials increased, the less subordinates headed for the hidden food first ($z = -2.91$; $P = 0.004$), and thus the more they headed for the visible food first. The hierarchical rank difference had no effect on whether the subject chose between hidden and visible food ($z = -0.27$; $P > 0.05$). Regarding only the two first trials per dyad, subordinates headed for hidden food significantly more than visible food in conditions 1 (W = 54; T+ = 60; T− = −6; $P = 0.01$) and 2 (W = 104; T+ = 112; T− = −8; $P = 0.002$). When they saw both pieces of food, subordinates headed significantly more frequently towards the hidden food when they were released at the same time than when they were released before the dominant (Fig. 3; $z = -4.11$; $P < 0.0001$). Regardless of random effects, all models assessed for dominants' first movement direction towards food reported a variance for the random effects equal to zero (i.e. each random parameter in each GLMM is equal to zero). As these parameters are meant to deal with dependence

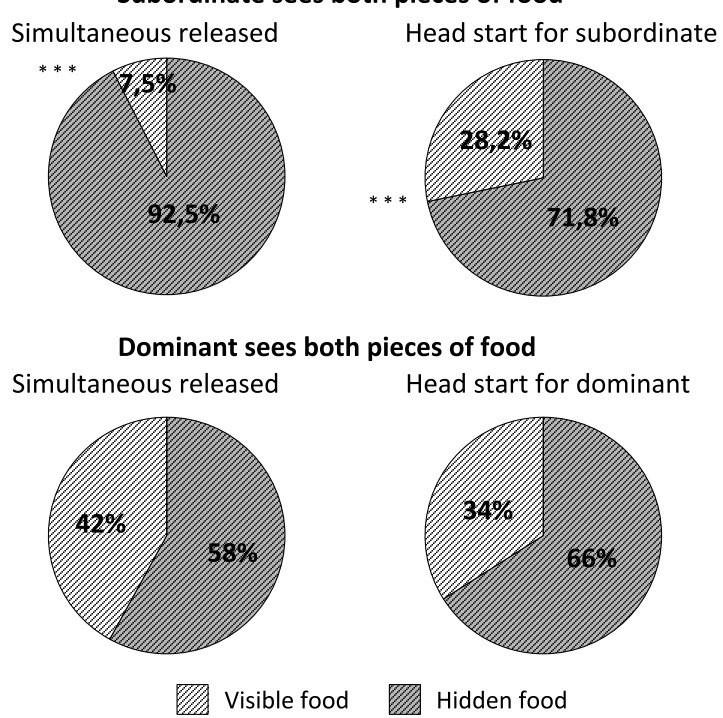

**Figure 3** First direction choice taken by the subject able to see both pieces of food, i.e. the subordinate in the conditions 1 and 2 and the dominant in the conditions 3 and 4, towards food (hidden or visible) in the different experimental conditions. *** $P < 0.0001$.

among repeated measurements, dominants' data could be considered as independent, allowing us to perform a GLM and not a GLMM that fit our data better (GLM4). When fitting the GLM, none of the explanatory variables had a significant effect and none of the probabilities for both directions differed significantly from 0.5. Thus, dominants showed no significant preference for moving towards hidden or visible food in the two conditions in which they had visual access to the two pieces of food (conditions 3 and 4; Fig. 3; $P > 0.05$). Regarding only the two first trials per dyad, dominants again showed no preference for hidden or visible food in both conditions 3 (W = 20; T+ = 24; T− = −4; $P > 0.05$) and 4 (W = 0; T+ = 52.5; T− = −52.5; $P > 0.05$) in their two first trials per dyad.

## Outcome of the test

The best models were those with the identity of the dominant as a random effect. Dominants obtained both pieces of food (outcome 1; GLMM5) significantly more when they saw both pieces of food and were released at the same time (condition 4: z = 8.01; $P < 0.001$) and also with a short head start on the subordinate (condition 3: z = 8.60; $P < 0.001$) than when they saw only one piece of food and were released at the same time as subordinates (condition 1; Fig. 4). Dominants obtained both pieces of food (outcome 1) significantly more when they saw both pieces of food and were released at the same time (condition 4: z = 8.27; $P < 0.001$) and also with a short head start on subordinates (condition 3: z = 8.79; $P < 0.001$) than when they saw only one piece of food

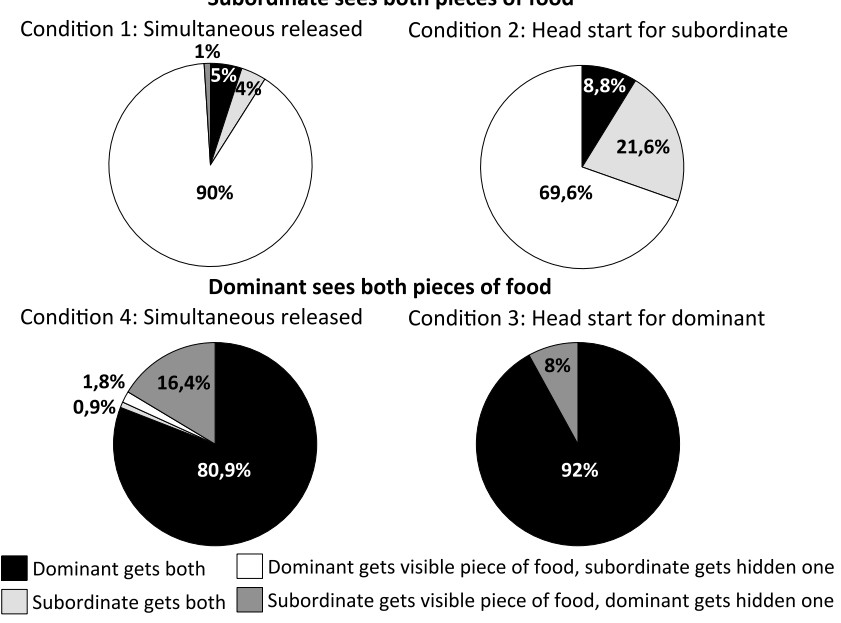

**Figure 4** Outcome of encounters in the different experimental conditions.

and were released after a short head start for subordinates (condition 2). Moreover, dominants tended to obtain both pieces of food more often in condition 3 than in condition 4 (z = −2.40; P = 0.073). The number of trials and the hierarchical rank difference between the tested individuals had no significant effect on the probability of outcome 1 (Fig. 4; P > 0.05).

Subordinates were significantly more likely to obtain both pieces of food (outcome 2; GLMM6) when they were released before the dominant (condition 2) than when they were released at the same time as the dominant (condition 1; z = 4.02; P < 0.0001; Fig. 4). Moreover, subordinates obtained the two pieces of food significantly more often with increasing trials (z = 2.64; P = 0.008). The hierarchical rank difference between the two individuals had no significant effect on the probability of outcome 2 (z = 0.95; P > 0.05).

Dominants got the visible food and the subordinate the hidden one (outcome 3; GLMM7) significantly more often when the subordinate saw both pieces of food and was simultaneously released (condition 1; Fig. 4) than when released with a short head start on the dominant (condition 2; z = −4.13; P < 0.0001). As the number of trials (z = −2.23; P = 0.03) and the hierarchical rank difference increased (z = −2.29; P = 0.02), the more likely was the dominant to retrieve the visible food and the subordinate the hidden one.

Occurrences of the subordinate getting the visible food and the dominant the hidden one (outcome 4; GLMM8) were significantly more frequent when the dominant saw both pieces of food and was released at the same time (condition 4; z = 2.23; P = 0.03) than when released before the subordinate (condition 3; Fig. 4). Neither number of trials nor the hierarchical rank difference had a significant effect on the probability of outcome 4 (P > 0.05).

**Table 3 Occurrences of first direction towards visible food item and occurrences of outcome 2 (Subordinate obtained two pieces of food) after subordinate chose to head for the visible piece of food in conditions 1 and 2.**

|  | Lady | Nema | Nenno | Nereis | Vishnu | Wotan | Yannick |
|---|---|---|---|---|---|---|---|
| Subordinate headed for the visible food | 15 | 1 | 8 | 0 | 1 | 0 | 15 |
| Subordinate headed for the visible food and obtained both pieces of food (outcome 2) | 9 | 0 | 6 | 0 | 0 | 0 | 8 |

Regarding the two first trials only, subordinates obtained the hidden piece of food rather than the visible one significantly more in conditions 1 (W = 66; T+ = 66; T− = 0; P = 0.001) and 2 (W = 153; T+ = 153; T− = 0; P < 0.0001). By contrast, dominants did not obtain one of the two pieces of food significantly more than the other in condition 3 (W = 4; T+ = 5; T− = −1,0; P > 0.05) or condition 4 (W = 8; T+ = 9; T− = −1,0; P > 0.05).

### Effect of first direction on the outcome

Three subordinates (Lady, Nenno and Yannick) of the seven tested obtained both pieces of food when they first headed for the visible one (Table 3). The best model concerning outcome 2 for observations made when subordinates headed first for visible food in conditions 1 and 2 was the one with no random effect (GLM9). This model reported a significant effect of first movement direction by subordinates on the likelihood of outcome 2: when subordinates first headed for visible food they obtained the two pieces significantly more than when they first headed for the hidden one (z = −6.72; P < 0.0001).

### DISCUSSION

We tested Tonkean macaques in an ecologically relevant competitive situation originally proposed by *Hare et al. (2000)* in which a subordinate subject could choose to retrieve a food item hidden from the view of a dominant conspecific and another one visible to both the dominant and itself. Our hypothesis was that if subordinates but not dominants preferentially went for the hidden food, this could be evidence of visual perspective taking in this macaque species. In agreement with previous studies on great apes (chimpanzees: *Hare et al., 2000*; *Hare, Call & Tomasello, 2001*; *Bräuer, Call & Tomasello, 2007*) and monkeys (rhesus macaques: *Flombaum & Santos, 2005*; long-tailed macaques: *Overduin-de Vries, Spruijt & Sterck, 2014*), our results provide support for this hypothesis in a tolerant macaque species: the Tonkean macaque. However, we cannot completely rule out an alternative explanation, namely that subordinates just avoided visible food potentially looked at by dominants throughout the transparent trapdoor.

Subordinates preferentially headed for food (hidden or visible) when they saw both pieces of food and were released simultaneously or with a short head start on the dominant, whereas dominants always went for food. This result validates the naturalistic experimental paradigm used here: placing subordinates in a food competition situation with dominants. In previous studies with chimpanzees (*Karin-D'Arcy & Povinelli, 2002*;

*Bräuer, Call & Tomasello, 2007*), marmosets (*Burkart & Heschl, 2007*) and capuchins (*Hare et al., 2003*), limited space probably caused subjects to engage in scramble competition rather than use perspective taking skills. The size and spatial arrangement of our testing area clearly induced a competitive situation appropriate for revealing perspective taking by subordinate macaques. Moreover, the greater the hierarchical rank difference between the two members of each was, the less the subordinate headed for food, further supporting the validity of the competitive situation.

It seems reasonable to suggest that the experimental context–collaborative or competitive–might have a lesser impact on a species with a more fluid social organization than species demonstrating a more strict social system. However, the evidence for this is equivocal. One the one hand, we previously failed to demonstrate that Tonkean macaques distinguish between open and closed eyes in a cooperative interspecific context (*Canteloup, Bovet & Meunier, 2015*). On the other hand, *Costes-Thiré et al. (2015a)* and *Costes-Thiré et al. (2015b)* found no evidence of auditory perspective taking in Tonkean macaques tested in a food competition situation with a human experimenter, or intention reading abilities when they had to cooperate for food with a human partner.

In the present study, when they saw both pieces of food subordinates preferentially headed for the hidden food when they were released simultaneously with the dominant. The same result emerged when subordinates were released slightly before the dominant, allowing them to choose independently of the dominant's choice. Indeed, when released at the same time, the subordinate could head for the hidden food because at the same time the dominance headed for the only food it could see, namely the visible one.

For this reason we ran more trials than is usual in these kinds of studies (e.g. *Hare et al., 2000*). This could lead to the criticism that, in our study, subordinates simply learned the best way to get food, requiring no perspective-taking ability. To deal with this explanation, we ran the same statistical analyses as *Hare et al. (2000)* on only the two first trials for each dyad, and found exactly the same results as obtained using all the data. These results suggest that, like other primate species studied (*Hare et al., 2000*; *Hare, Call & Tomasello, 2001*; *Bräuer, Call & Tomasello, 2007*; *Overduin-de Vries, Spruijt & Sterck, 2014*), Tonkean macaques appear capable of understanding what a conspecific can and cannot see. This conclusion is strengthened by the fact that dominants showed no preference for either the hidden or visible food. This finding rules out the possibility that macaques have a general preference for hidden food. However, a final alternative explanation invoking behavior reading rather than perspective taking may be proposed. According to the "evil eye hypothesis" (*Kaminski, Call & Tomasello, 2008*) and other authors' "behavioral rules" accounts (*Heyes, 1998*; *Povinelli & Vonk, 2003*; *Povinelli & Vonk, 2004*), the subordinates' behavior could be influenced by what it sees the dominant doing through the transparent trapdoor. In other words, subordinates might prefer to head for the hidden food because the visible one will have been watched by the "evil eye" of the dominant, and thus considered as "contaminated." They might also have learned that food coveted by the dominant is out of bounds. Unfortunately, our experimental design does not allow us to refute this alternative explanation; the use of

one-way mirrors as in *Overduin-de Vries, Spruijt & Sterck's (2014)* study on long-tailed macaques might be an appropriate procedure that would allow to rule out this low-level explanation.

Given that visual perspective taking has been demonstrated in a phylogenetically very close species, the long-tailed macaque (*Overduin-de Vries, Spruijt & Sterck, 2014*), we think it likely that this ability has evolved as common trait shared by macaque species. Some of the individual strategies that we observed further support our view that Tonkean macaques are capable of visual perspective taking. One such strategy used by subordinates consisted of traversing and leaving the testing area, and if followed by the dominant, rapidly doubling back to retrieve the hidden food (see Video S3). Another strategy was to head first for the visible food. Indeed, as the number of trials increased, the less the subordinates headed for the hidden food first. Moreover, when they saw both pieces of food, subordinates headed for hidden food significantly less often when released with a short head start on the dominant than when released simultaneously. Surprisingly, when released before the dominant, subordinates moved first to the visible food in almost thirty percent of cases. In terms of the outcome of the test, in this condition, subordinates often managed to obtain both pieces of food when they first headed for the visible food, and this occurred increasingly frequently over repeated trials. On the basis of these observations we conclude that three of the seven subordinates tested adopted this alternative strategy to get both pieces of food by taking into account what the dominant could and could not see. This kind of strategy was recently reported in a more despotic macaque species: the long-tailed macaque (*Overduin-de Vries, Spruijt & Sterck, 2014*), and similar behavior was reported in Tonkean macaques by *Ducoing & Thierry (2003)*. In the latter study, subordinates stopped approaching a hidden fruit, avoided being followed, or took a wrong direction when monitored by a dominant unaware of the location of the food. Together, these observations raise questions about the mechanisms underlying these strategies. Do they rely on relatively simple cognitive processes, such as withholding information due to behavioral inhibition caused by the presence of the dominant, or are more complex cognitive processes involved, such as tactical deception? In other words, do Tonkean macaques learn by operant conditioning to anticipate the consequences of their acts on the behaviors of others or/and do they intentionally attempt to manipulate the knowledge of others? Further investigations are necessary to be able to answer these unresolved questions.

To conclude, our experiment adds to the growing literature on components of Theory of Mind in nonhuman primates, especially in monkeys and notably in macaques (e.g. *Flombaum & Santos, 2005*; *Santos, Nissen & Ferrugia, 2006*; *Costes-Thiré et al., 2015a*; *Costes-Thiré et al., 2015b*; *Overduin-de Vries, Spruijt & Sterck, 2014*). Like chimpanzees (*Bräuer, Call & Tomasello, 2007*; *Hare et al., 2000*; *Hare, Call & Tomasello, 2001*; *Krachun & Call, 2009*) and despotic Old World monkeys as rhesus (*Flombaum & Santos, 2005*) and long-tailed macaques (*Overduin-de Vries, Spruijt & Sterck, 2014*), Tonkean macaques, socially more tolerant, seem capable of level 1 of visual perspective taking, although further experiments are needed to rule out a low level alternative explanation. Additionally, we observed an alternative strategy developed by subordinates to obtain all

the available food. These findings show the advantages of a naturalistic experimental paradigm that recreates socio-cognitive problems that nonhuman primates face in their natural environment.

## ACKNOWLEDGEMENTS

The authors are grateful to Nicolas Herrenschmidt, Yves Larmet and the whole team of the Centre de Primatologie de l'Université de Strasbourg for allowing them to conduct this study. The authors are particularly thankful to Steve Lapp, Adrien Panter and Jean-Marc Woock for building the experimental apparatus and for their technical assistance. James R. Anderson and Nailah Ford Burrell are warmly thanked for editing the English of the manuscript. Finally we thank Juliane Kaminski, Academic Editor, and the two anonymous referees for their valuable comments on the manuscript.

### Funding

The research in this paper was supported by the Centre de Primatologie de l'Université de Strasbourg and the University of Strasbourg Institute for Advanced Study (USIAS) as part of a USIAS Fellowship. CC also received a grant "Expériences de jeunes" for innovative projects from the Alsace Region for her work. The funders had no role in study design, data collection and analysis, decision to publish, or preparation of the manuscript.

### Competing Interests

The authors declare that they have no competing interests.

### Author Contributions

- Charlotte Canteloup conceived and designed the experiments, analyzed the data, contributed reagents/materials/analysis tools, wrote the paper, prepared figures and/or tables, reviewed drafts of the paper.
- Emilie Piraux performed the experiments.
- Nicolas Poulin analyzed the data, contributed reagents/materials/analysis tools, reviewed drafts of the paper.
- Hélène Meunier conceived and designed the experiments, analyzed the data, contributed reagents/materials/analysis tools, reviewed drafts of the paper.

### Animal Ethics

The following information was supplied relating to ethical approvals (i.e., approving body and any reference numbers):

The procedure adheres to the French legal requirements for the Use of Animals in Research. This experiment was approved by the Animal Experiment Committee of the Centre de Primatologie de l'Université de Strasbourg and by the CREMEAS ethics committee (approval for conducting experiments on primates n° AL/46/53/02/13).
## Data Deposition

Raw data are available in the Supplemental Information.

## Supplemental Information

Supplemental information for this article can be found online at http://dx.doi.org/10.7717/peerj.1693#supplemental-information.

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
