# Peer review of "Do Tonkean macaques (Macaca tonkeana) perceive what conspecifics do and do not see?"

_PeerJ, doi:10.7717/peerj.1693_

## Round 0.1 · original submission · Major Revisions

· Academic Editor

Major Revisions

I agree with both reviewers comments. You will need to revise the manuscript in line with their comments. Please have a native speaker proof read the manuscript as I agree with reviewer 1 quality of the writing needs to improve and the English corrected. The manuscript is partly difficult to understand (especially the methods section) and therefore it is difficult to review the paper.

Reviewer 1 ·

Basic reporting

I hate to start with this because I am certainly sympathetic to the additional burden faced by non-native English speakers who must write scientific manuscripts in English, but, unfortunately, my ability to evaluate a number of features of the study is impeded in some cases by the quality of the writing. Currently, many concepts are expressed in very clunky ways, sometimes making it unclear what the authors are trying to communicate. In some cases, I had to read a sentence three times before I felt that I could even make a best guess at what the authors intended to communicate and often these were critical sentences about important aspects of the design, exclusion criteria, or analyses. Consequently, without further improvement of the writing—aimed at enhancing clarity and ease of reading—it will be difficult to fully evaluate the authors’ work. Consequently, I strongly recommend editing by a native or more experienced English speaker for clarity, diction, grammar, and general flow.

Experimental design

Details of the experimental design remain unclear (see General Comments for the Author).

Validity of the findings

Details of the data and analyses remain unclear (see General Comments for the Author).

Additional comments

The authors have performed an experiment on visual perspective taking in Tonkean macaques. This experiment is modeled after a pioneering study by Hare and colleagues (2000), which provided the first strong evidence that a nonhuman species (chimpanzees) could reason about what others can and cannot perceive. Subsequent work on understanding of others’ mental states, or theory of mind, has also largely focused on chimpanzees. Therefore, any work in other species, including more distant relatives of humans, is highly valuable as it can elucidate the phylogenetic distribution of theory of mind abilities among primates. Since most previous work has tested despotic species (chimpanzees and rhesus macaques) and relied on competitive paradigms, Tonkean macaques, as a more socially tolerant species, are well positioned to contribute key comparative data that addresses questions like: what ecological and social pressures favor the evolution of theory of mind abilities, and what contextual features motivate species of different ecological and social backgrounds to exhibit social cognitive skills? For these reasons, I believe that the submitted manuscript holds the potential for considerable merit. However, it is not yet fit for publication and I offer below a number of recommendations to improve the manuscript.

MAJOR POINTS:
Recommended Changes:
To Analyses:
One way that this study differs significantly from earlier work on great apes is that a very few subjects contribute most of the data. For example, 83% of the trials involve one of two subordinates, each of which participated in more than 150 trials. I think that the paper would be strengthened if you can show, like Hare et al (2000), that during the first couple trials (e.g., the first 2 trials per dyad per condition, like Hare et al (2000)) you get the same effect as with the overall dataset. Otherwise, it’s completely unclear whether this is a spontaneous effect or if subjects learned the best strategies over time. For a more direct comparison, it would also be nice to see results replicated with this small sample size using the same statistical techniques as Hare et al (2000).

Lines 418-426: The authors need to be clearer in explaining the alternative strategy that the macaques are taking. Importantly, however, they have not specifically shown that subordinates are more likely to get both pieces of food when they first target the visible food than when they first target the hidden one. This is a necessary analysis to demonstrate the alternative strategy that the authors are alluding to. Additionally, the authors should test whether subordinates or dominants get more pieces of food when first approaching the hidden food versus the visible food. This would tell them whether the strategy of first targeting the visible food is consistently successful at maximizing overall food acquisition, or whether it is just more likely (than first targeting the hidden food) to produce the unlikely outcome of acquiring both pieces of food.

To Writing/Presentation:
The authors need to outline a series of hypotheses and predictions that justify the design of their experiment. Obviously the design is generally based off of Hare et al (2000, 2001); however, it is not identical. Therefore, the authors should make clear which elements from Hare et al (2000, 2001) are included and which are not and justify these choices. Between the two papers, Hare and colleagues (2000; 2001) performed nearly ten experiments, each of which involved several conditions. Why are the authors’ chosen conditions the critical ones and what about all of the controls that Hare and colleagues did? What is the purpose of each condition? (What does each condition demonstrate or control for?) Why were the chosen analyses pursued? How did those analyses test the predictions of the hypotheses that guided the study?

The introduction should be much more targeted. Currently too much time is spent on providing background that is not terribly relevant to the studies at hand. This is in line with the above comment on framing the paper in terms of a major question and clear hypotheses.

More justification should be provided for the value of specifically testing Tonkean macaques (what contributions do they make to our understanding of the phylogenic distribution of ToM abilities? What can we learn from comparisons between more and less tolerant species about the evolution of ToM or the contexts in which it is expressed?).

The discussion must address alternative explanations to mind-reading and why they are unlikely.

Why don’t dominants show a preference for the visible food in conditions in which they could only see the visible food, especially when they and the subordinate were released at the same time?

New tables displaying all properties and results of the final models should be created and included in the supplement.


Methodological Clarifications:
Line 206-209: Using a binomial test? How did you determine when to stop trials and test for significance? Why was the trial number different between pairs?

In what way was testing opportunistic? Why was the number of trials per dyad variable? How many trials were involved in a session? Did each session involve only a single condition or did subjects participate in trials of all condition types within the same session? Were the food locations counterbalanced within each dyad? Within each condition? Within each session?

What was the baiting procedure that left one animal ignorant to one food’s location and the other aware of both? Are the trapdoors opaque? Did either animal witness any of the baiting, or were they both ignorant until the doors were raised? How did they know that another animal was present in the third room? It is important to be clear about what information the decision-maker has at hand? What did they see that might have allowed them to reason about the mental states of their competitor, or respond to his behavior? In cases where they could respond to behavior (e.g., gaze or intimidation), were controls implemented?

How do subjects know that they are about to compete?

Lines 249-251: This sentence is very confusing but I believe what you mean to say is: “In some cases, subjects eventually realized that two pieces of food were present in every trial and began searching both locations, even on trials in which they only had visual access to one piece of food. We excluded these trials.” How long did it take before subjects learned that food was present in two locations? Did you stop testing animals (in their current dyad or any other dyad) once it became apparent that they were aware that food was present in two locations? Did you intersperse sessions with any other kinds of trials so that subjects could not always be sure how many pieces of food were present and where they were located?

Lines 301-303: I don’t understand what this means. What do you mean by “more the hierarchical rank difference was important?” Are you reporting an interaction effect? Did your models include any interaction effects?

Where are the descriptive stats? Figures 2-4 are useful, but I would like to know the number of data points involved in each condition as well as the number of subjects. Also, since the authors run analyses on how condition impacts outcome, there should be descriptive stats on those datasets (for each outcome, the number of trials per subject per condition)

How many data points went into each model? How many subjects were involved in each model?

Did models just include as a fixed effect trial number for the dyad, or did they also include trial number for the subject? It’s important to control for subjects’ overall learning throughout the experiment, since it’s clear that they are gaining experience with the task throughout.

When you chose the model to report in each case, did you simply choose the lowest AIC option or did you make sure that the lowest AIC option had a significantly better fit or was at least 2 AIC points below the next best fitting model? If you are using a model selection procedure to identify the best fit model, you should report all models within 2 AIC units of the lowest AIC model or use model averaging techniques. For further guidelines on model selection techniques, see Burnham et al (2011) “AIC model selection and multimodel inference in behavioral ecology: some background, observations, and comparisons.”



MINOR POINTS:
There are some cases of run-on sentences that should be broken down into two or more sentences (e.g., lines 45-49)

Lines 49-51: Byrne and Whiten, 1988 is an inappropriate citation since what they report is anecdotes of deception in nonhuman primates. Only experiments can implicate higher order cognition.

Lines 85-92: There is evidence of level 2 perspective taking in 3 year old human infants (Moll and Meltzoff, 2011)

Lines 127-129: Hare et al (2000) and Brauer et al (2007) also used a headstart procedure

Lines 139-142: Also cite Sandel et al (2011) “Evidence from four lemur
species that ringtailed lemur social cognition converges with that of haplorhine primates” and MacLean et al (2013) “Group Size Predicts Social but Not Nonsocial Cognition in Lemurs”

Line 322: Do you mean random effects?

Line 323: Please explain more clearly and thoroughly why these data should be considered independent.

Line 342: Should it say, “were released after a short head start for subordinates” ?

Lines 386-391: The authors should first explain that Brauer et al (2007) found that spatial features of the setup influence subjects’ behavior and that proximity of food to the dominant seems critical for ensuring that subjects use perspective taking skills, rather than a scramble/agility strategy, to maximize food acquisition. Then they can say that, unlike previous studies that failed to provide evidence of perspective taking, their setup involved this optimal spatial arrangement, which likely contributed to their success. Currently, it would be confusing for readers who are not highly familiar with Brauer and colleagues’ (2007) findings.

Lines 436-439: At the very least you should acknowledge the work on rhesus macaque theory of mind by Laurie Santos and others.


Final comment:
I really want this paper to become publishable because I think it holds real theoretical importance, but substantial revisions are currently required before it will reach that state.

Reviewer 2 ·

Basic reporting

The authors tested Tonkean macaques in a food competition paradigm to assess visual perspective taking. They found that subordinate individuals preferentially chose a piece of food not visible to the dominant individual, even under head-start conditions, which is consistent with level one perspective taking. Furthermore, they report that over time, some individuals started to use deceptive strategies. These results are interesting, but there are two important issues that have to be solved prior to publication. The first issue concerns the theoretical embedding and consequently the interpretation of the results (see section on “experimental design”); the second issue concerns the sample size. 90% of all trials in the crucial condition where a subordinate individual sees two pieces of food are from only two focal individuals, all trials are from only four individuals (no headstart condition; headstart condition: 72% of all trials are from 2 focal individuals only, and all trials from a total of 7 individuals; see also section on the validity of the findings).

Experimental design

The authors use the competitive paradigm developed by Hare et al. and suggest that it has been successfully solved via perspective taking by apes and longtailed macaques, but not by New World monkeys. Therefore, the data from the Tonkean macaques would support the idea that the trait emerged at the evolutionary split between New and Old World monkeys. This is not entirely correct.
When summarizing the previous findings, it is important to make a distinction between the behavioral criterion of passing the task (i.e. that subordinates consistently choose the hidden food when they have a headstart) and the underlying mechanism (behavior reading or perspective taking).
In the past, chimpanzees, longtailed macaques and common marmosets have clearly passed this task, but capuchin monkeys did not. The underlying mechanism is more difficult to identify. In longtailed macaques, perspective taking is most likely since one way mirrors had been used to make the behavior reading explanation unlikely (Overduin de Vries et al. 2104). For marmosets, a potential behavior reading mechanism had been pointed out. However, the existence of such a mechanism does by no way exclude that they nevertheless do understand level 1 visual perspective. More importantly, the same mechanistic alternative mechanism could likewise explain the chimpanzee behavior, as well as the Tonkean behavior.
The current findings are thus reported by interpreting the same behavior (i.e. passing the competition task) differently in different species, i.e. as representing visual perspective taking in chimpanzees and Tonkean macaques, but as representing behavior reading in marmosets. This is not acceptable.

Validity of the findings

To account for the unequal distribution of trials per focal individual and dyad, the authors include the random effects of individual, partner and dyad in the model. However, whenever a model without these random effect provides a lower AIC, they exclude the random effect from the model. This procedure is biologically not meaningful, because measuring the same individual more than once remains a pseudo-replication regardless of whether variation within this individual is higher or lower compared to the variation between individuals. This is particularly relevant because in the crucial condition, where the subordinate could see both pieces of food, 90% of all trials are from the same focal individual (no headstart condition, all trials are from only four different subordinate subjects; headstart condition: 72% of all trials are from 2 focal individuals, and all trials from a total of 7 individuals. These figures are based on Table 2, under the assumption that trials that were excluded from the analysis are not listed there). I think this is not sufficient for a publication and suggest to add data from more focal individuals. In fact, when adding additional individuals, it would be most useful to adopt the procedure used by Overduin de Vries et al. (201ç)
The finding that after repeated testing, subordinates start to use deceptive strategies thus seems to apply to these two individuals only. If so, the data for these two individuals should be presented separately for each individual over time.

Additional comments

Some minor comments:
Line 100: note that marmosets are able to use human gaze cues, with high precision and including eyes only cues (Burkart & Heschl 2006, rather than 2007)
Line 156: what are these insights about the influence of sociocultural environment on perception reading abilities? This is an interesting topic but it would need to be elaborated a bit more.
Line 249-255: are these trials included or excluded in Table 2?
Line 277: what was the response in the GLMMs separate for each outcome?
Line 314-315: does this apply to the simultaneous or to the headstart condition?
Line 386-390: Why would one expect that the measures have to be the same for capuchins, marmosets and chimpanzees??

---

## Round 0.2 · Minor Revisions

· Academic Editor

Minor Revisions

You can see that the reviewer thinks that your manuscript has greatly improved and after reading your revised manuscript I agree. I do, however, still have some few comments, below, which I would like you to address.

1. In the abstract (and also later on) you talk about Perspective taking as a “prerequisite of ToM”. I think the term “prerequisite” is a bit odd in that context and it would be good if you could find a different way of describing the connection of perspective taking and ToM. In some sense ToM is a set of skills one of which is perspective taking, however, having perspective taking skills does not necessarily mean there is evidence for ToM

2. In the abstract (Line 26) add the species name.

3. Line 83: I think it would be good to add a reference for the statement made here.

4. Line 84: saying “with a human model” is a bit of an odd description. Could you say “following the gaze of a human experimenter” instead?

5. In the introduction (line 270 onwards) it would be important to describe some of the control conditions that Hare et al. 2000 conducted in order to rule out alternative explanations but perspective taking.

6. Line 330 onwards: I am not sure this part is entirely clear. What would be a possible ‘behaviour reading explanation” and how was it ruled out. I think the reader would benefit from more information here.

7. Line 342 onwards: your are citing Hare et al. 2000 but then talking about ‘monkeys” so are you actually referring to Hare et al. 2003? Would be good if you could clarify.

8. Line 390 onwards: I think it is great that you now included more information about the species but I think that should appear earlier in the introduction and could still be extended a bit. Why might it be especially interesting to compare the data of a more tolerant species to the data we already have from more despotic species? I think it would be great if you could elaborate on that a bit more.

9. One comment on your Method section: I think for the reader it would be easier if the conditions had names instead of just being numbered e.g. Hidden-Visible condition like in the original Hare et al. studies.

10. Line 536 onwards: this is not entirely clear. What do you mean by “at random per dyad”. Were trials presented in sessions? How were they presented. Could you explain a bit more?

11 Line 613 onwards: whats the exact criterion for excluding trials? Was there a clear definition? This needs a little more explanation. Also, was there reliability coding for the exclusion of trials also? And it seems the reliability coder was not blind to the study right? This should be mentioned somewhere.

12. Overall comment: As you cannot rule out alternative explanations e.g., the subordinate might have reacted on seeing (through the transparent door) the dominant orient towards one piece of food, I think you could tone down your interpretation even more. You are discussing these options but I think you could be even more cautious in your interpretation.

Reviewer 2 ·

Basic reporting

Overall, the manuscript has much improved and most of my concerns have been addressed. I am not entirely satisfied by the answer regarding the random effects, namely that it “has been deeply discussed between the authors, biologists and statistician” and that their “conclusion converged”. The fact remains that the results are based on a very low number of subjects, and I don’t see from their answer how it could be biologically meaningful to remove focal individuals as random effects. Priority should be given to run balanced experiments with sufficient observations in all conditions, rather than relying on GLMM’s to fix incomplete experimental schedules. However, the newly added analysis of the first two trials shows that the results are stable, and I recommend to publish the paper. As a final recommendation, I would suggest to add the alternative strategies suggestive of perspective taking in the abstract because otherwise, many people will miss this important piece of evidence.

Experimental design

no comment

Validity of the findings

no comment

Additional comments

no comment

---

## Round 0.3 · accepted · Accept

· Academic Editor

Accept

I think you have addressed all comments and toned the manuscript down even more. I agree that specifying the names for the conditions might not make things easier and therefore just numbering the conditions is probably the best option.